# Somatic Embryogenesis and *Agrobacterium*-Mediated Gene Transfer Procedures in Chilean Temperate Japonica Rice Varieties for Precision Breeding

**DOI:** 10.3390/plants13030416

**Published:** 2024-01-31

**Authors:** Marion Barrera, Blanca Olmedo, Carolina Zúñiga, Mario Cepeda, Felipe Olivares, Ricardo Vergara, Karla Cordero-Lara, Humberto Prieto

**Affiliations:** 1Natural Sciences, Mathematics, and Environment Faculty, Metropolitan Technological University, Santiago 8330526, Chile; marion.barrerac@utem.cl; 2Biotechnology Laboratory, La Platina Research Station, INIA-Chile, Santiago 8831314, Chile; bolmedo@inia.cl (B.O.); karolina.z.c@gmail.com (C.Z.); mario.cepeda@usach.cl (M.C.); felipe.olivares@inia.cl (F.O.); ricardo.a.vergara.g@gmail.com (R.V.); 3Rice Breeding Program, Quilamapu Research Station, INIA-Chile, Chillán 3780000, Chile; kcordero@inia.cl

**Keywords:** genetic transformation, geminivirus-derived vector, cold-adapted rice

## Abstract

Rice (*Oryza sativa*) varieties are generated through breeding programs focused on local requirements. In Chile, the southernmost rice producer, rice productivity relies on the use and generation of temperate japonica germplasms, which need to be adapted to the intensifying effects of climate change. Advanced biotechnological tools can contribute to these breeding programs; new technologies associated with precision breeding, including gene editing, rely on procedures such as regeneration and gene transfer. In this study, the local rice varieties Platino, Cuarzo, Esmeralda, and Zafiro were evaluated for somatic embryogenesis potential using a process that involved the combined use of auxins and cytokinins. An auxin-based (2,4-D) general medium (2N6) allowed for the induction of embryogenic masses in all the genotypes. After induction, masses required culturing either in N6R (kinetin; Platino) or N6RN (BAP, kinetin, IBA, and 2,4-D; Cuarzo, Esmeralda, and Zafiro) to yield whole plants using regeneration medium (N6F, no hormone). The sprouting rates indicated Platino as the most responsive genotype; for this reason, this variety was evaluated for gene transfer. Fifteen-day-old embryo masses were assayed for *Agrobacterium*-mediated transformation using the bacterial strain EHA105 harboring pFLC-*Myb*/*HPT*/*GFP*, a modified T-DNA vector harboring a geminivirus-derived replicon. The vector included the green fluorescent protein reporter gene, allowing for continuous traceability. Reporter mRNA was produced as early as 3 d after agroinfiltration, and stable expression of the protein was observed along the complete process. These achievements enable further biotechnological steps in these and other genotypes from our breeding program.

## 1. Introduction

Rice (*Oryza sativa* L.) is one of the world’s most important food crops and is a staple for more than half of the world’s population. Plant breeding programs and the development of new genotypes with increased fitness and productive capabilities are currently a priority in an era conditioned by the effects of global climate change [1]. Chile, the southernmost rice producer, has approximately 27,000 ha under rice cultivation, which yields an average of 6.5 tons/ha; production is exclusively based on the temperate *japonica* varieties [2]. Cultivar choice follows the weather conditions in agricultural areas, which include spring cold during flowering as well as water restrictions and heat shock events during harvest in summer.

Recent advances in precision breeding, including gene editing (GE), enable the targeted and precise genetic modification of the rice genome [3]. The implementation of these technologies has led to a renewed interest in optimizing plant transformation and regeneration techniques, including *Agrobacterium tumefaciens*-mediated gene transfer. In most studies of gene transfer in rice, researchers have used either immature embryos or embryogenic calluses in the starting procedures [4]. The former involves the use of controlled conditions at the time of seed generation; the latter involves mature seeds and is more affordable. Somatic embryogenesis (SE) is a powerful tool for dedifferentiation and subsequent plant regeneration, having strongly impacted important rice genotypes [5,6] and consolidated protocols [4]. However, SE induction is a genotype-dependent response, which is a bottleneck to overcome. *Japonica* cultivars such as Nipponbare are highly responsive to SE induction and have transformable calli [7]. Conversely, many *indica* [8], tropical *japonica* [9], and locally generated genotypes [10] require dedicated procedures for tissue culture. *Agrobacterium*-mediated gene transfer methods are relatively easy to incorporate into SE-based regeneration procedures, leading to successful plant genetic engineering [11]. In addition, modified T-DNA containing autonomously replicating viral modules can be used to create high-efficiency expression vectors of different sequences of interest [12], including GE machinery [13,14,15].

Spring-cold-adapted *japonica* varieties are a priority for local breeding programs. Producing climate-resilient genotypes contesting biotic and abiotic stresses is an active aim for biotechnology, allowing for an accelerated genetic gain for important genotypes. Thus, the development of technical procedures associated with both SE and gene transfer techniques for these materials would be impactful. Here, we describe the implementation of SE and *Agrobacterium*-mediated gene transfer procedures for four relevant local genotypes.

## 2. Results

Isolated embryos (7 d after germination; Figure 1a) and the cultivation procedures using 2N6 medium (2,4-D) led to SE induction (Figure 1a, “Induction”) and adequate calli multiplication and propagation in all the evaluated genotypes (Figure 1a, “Proliferation”). The results of a microscopy evaluation of cultured explants showed the induction of proembryogenic masses that efficiently achieved proliferative stages in which somatic embryos had multiplied. Globular cell masses formed as early as 15 d of cultivation, which then efficiently propagated on average 30 d after culture initiation (Figure 1b). These embryo cultures were successfully maintained via subculturing them every 15 d in this medium.

Differential responses among the assayed genotypes to culture conditions were observed when more advanced embryo stages leading to germination were induced. For embryo germination, two different media were evaluated: N6R (kinetin) and N6RN (kinetin, BAP, IBA, 2,4-D) (Figure 1b, “Germination”). Different average responses were observed among the genotypes (Figure 1c), but these germination rates were not associated with the final plant generation. Medium N6R led to the best sprouting in Platino and resulted in a high germination rate for Esmeralda and Zafiro; however, only Platino explants produced whole plants after culturing in N6F (no hormone) medium. Conversely, N6RN was more effective for the germination of Cuarzo, but this medium allowed for the whole-plant induction of Esmeralda, Zafiro, and Cuarzo after moving these explants from this medium to N6F. The final procedures for each of the evaluated genotypes are shown in Figure 1b.

The *Agrobacterium*-mediated gene transfer ability to somatic embryos was first evaluated using strains LBA4404, GV3101, and EHA105. We used the pFLC-*Myb*/*HPT*/*GFP* plasmid (Figure 2a). The results of the assays showed that early-developed embryos, generated 15 d after induction (Figure 1a; end of the “Induction” phase), were the most receptive cell type for agroinfection, which was exclusively verified only with the EHA105 strain (Figure 2b). The results for these infections, judged as GFP-emitting explants, are shown in Figure 2c (Appendix A) for all the genotypes. Here, Platino and Esmeralda showed higher percentages of *GFP*-expressing explants. In Platino, the most responsive genotype, transformed embryos were observed as early as 3 d after infection (Figure 2d). These explants exhibited continuous development under hygromycin-selective conditions (Figure 2e) up to active budding (Figure 2f, left). The time at which the selective agent was removed is shown in Figure 2f (right).

## 3. Discussion

Somatic embryogenesis is a unique process, valuable as a tool to enhance the genetic improvement in crop species when integrated with classical breeding programs and molecular biology techniques [16,17]. First developed using leaves [18], specific SE protocols are needed for different genotypes [19]. The combined use of auxins and cytokinins is an important factor in avoiding recalcitrance in monocots [20]. Here, SE was induced by culturing the calli produced from mature seeds using an auxin-based (2,4-D) general medium (2N6). We needed to adapt two different media for whole-plant production: (a) N6R medium, which contained only cytokinin (kinetin); and (b) N6RN, which included both cytokinins (BAP and kinetin) and auxins (IBA and 2,4-D). Plant regeneration was not achieved with the use of a single medium, whereas whole-plant generation was achieved for Platino after culturing in N6R. The other three varieties required N6RN for whole-plant generation.

As rice is a monocot model system and is an active focus of genome engineering technologies, rice improvements can directly achieve agronomical benefits [21]. SE has had technological impacts on clonal propagation, the production of synthetic seeds, and, as the goal here, gene transfer. Monocots are not the natural hosts of *Agrobacterium* species; thus, transformation can be extremely tedious [22]. As such, the development of a rice genotype with high embryogenic competence and determining its specific developmental stage for transformation is challenging [22]. Factors affecting the results of *Agrobacterium*-mediated transformation in monocotyledonous plants include (a) explants (i.e., genotype, type, and stage of explant), (b) gene transfer system (i.e., strain of *Agrobacterium* and type of vectors), and (c) the experimental conditions (i.e., acetosyringone, cocultivation temperature, and plant regeneration efficiency) [23]. For temperate *japonica* rice genotypes, these developments require previous knowledge and technique refinement [10]; for this, different media for culture establishment and regeneration, which affect the development and recovery of transgenic rice, have been described [24], which depend on the genotype. Our results showed differential responses of different rice varieties to culture treatments not only at this level but also at advanced stages involving rooting (under N6F cultivation), thus reinforcing the idea that general responses to SE and whole-plant generation are highly genotype dependent in this species. As such, different cultivars must be tested [25]. The variations in SE induction responsiveness and regeneration efficiency among different rice cultivars have established some differences between *indica* and *japonica* varieties [19], which can be attributed to different common wild ancestral populations [26]. The comparative transcriptome profiling between Nipponbare (*japonica*) and PB-1 (*indica*) showed that transcripts of functional importance are dynamically and differentially expressed among SE developmental stages and between the subspecies [19]. These results suggest an important participation of meristem development regulators and phytohormone signaling pathways essentially during proembryogenic callus induction, which may be responsible for stronger regeneration and differentiation of somatic embryos in *japonica*.

Contemporary breeding programs aim to produce new genotypes that can better adapt to the intensifying effects of climate change and environmental anomalies [21,23]. This need has encouraged researchers to expand the breeding repertoire to include complementary tools such as GE [12]; the procedures described here enable the use of these tools in these and additional local varieties to produce food in a region classified as highly vulnerable to climate change [27].

## 4. Materials and Methods

### 4.1. Embryogenic Callus Induction

Seeds from the varieties Platino, Cuarzo, Esmeralda, and Zafiro were hulled and washed for 5 min with running water and neutral detergent and then rinsed three times with water for 2 min. Washed seeds were disinfected for 1 min with ethanol 76% and rinsed three times for 2 min with sterile water. Seeds were transferred into a bleach solution (20%) supplemented with two drops of Tween 20, immersed for 20 min with regular stirring (80 rpm), and, finally, washed with sterile water six times over 2 min. Seeds were dried using filter paper. Dried seeds were allowed to germinate via cultivation for 3 d in Petri dishes with 2N6 medium [4] (Appendix A) at 27 ± 1 °C and with a 16 h/8 h (light/dark) photoperiod. Embryos were removed and cultured in the same medium for callus induction.

### 4.2. Somatic Embryo and Plantlet Production

The base methodology for our SE procedures was adapted from procedures reported by Hiei and Komari [4]. Briefly, induced proembryogenic calli were produced by culturing isolated embryos in 2N6 medium for 14 d in darkness and then under a 16 h/8 h photoperiod, as described above, both at 27 ± 1 °C. Explants were maintained in these conditions and subcultured every 15 d in the same medium up to somatic embryo formation. Somatic embryo sprouting was achieved through culturing somatic embryos (typically generated after 30 d from culture starting) in N6R medium [4] and N6RN medium (Appendix A). Budding points were isolated and cultured in an N6F medium (Appendix A) for rooting and whole-plant formation.

### 4.3. Geminivirus-Based T-DNA Vector

We constructed a universal fluorescent pLSL-cloning plasmid (pFLC-U), a modified T-DNA vector. The plasmid uses the pLSLGFP.R plasmid (Addgene #51501) [13] as a backbone, in which the fragment between *Xho*I and *Pac*I restriction sites (i.e., *GFP* and *HPT* gene cassettes) was replaced for a fragment containing two cassettes: (a) *GFP* expression cassette and (b) a recombination module. The *GFP* cassette was flanked by *Bsa*I restriction sites and consisted of the *Cestrum yellow leaf curling virus* (*CmYLCV*) promoter, the *GFP* gene, and the *Arabidopsis Heat shock protein* 18.2 (*HSP*) gene terminator. Contiguous to this cassette, we included the recombination module comprising the pGWB502 Gateway Recombination Cassette [28]. This replacing fragment was synthesized at Genewiz (Azenta Life Sciences, Burlington, MA, USA). The Gateway Recombination Cassette was included for future insertion of any sequence of interest. In this study, a particular pFLC-U version (i.e., pFLC-*Myb*/*HPT*/*GFP*) was constructed via a LR recombination reaction to clone the potato *StMybA*1 gene [29]. The golden gate reaction [30] was used to replace the *GFP* gene via *HPT-GFP* gene fusion (obtained from pGWB505, Addegene #74847) [30] using the *Bsa*I restriction site and T4 Ligase (New England Biolabs, Ipswich, MA, USA).

### 4.4. Agrobacterium-Mediated Transformation of Somatic Embryos

*Agrobacterium tumefaciens* EHA105 harboring pFLC-*Myb*/*HPT*/*GFP* was prepared via culturing the bacteria in 30 mL of LB medium (ThermoFisher Scientific, Waltham, MA, USA) supplemented with kanamycin (100 mg/L; MilliporeSigma, Burlington, MA, USA) for 24–36 h up to an OD_600_ of between 0.3 and 1.0. Bacterial cultures were centrifuged at 4300 rpm for 10 min, and the pellet was resuspended in PIMII medium [31] (Appendix A) supplemented with 100 μM acetosyringone (MilliporeSigma); the cultures were then incubated for 30 min at 100 rpm before embryo infection. Immature embryos generated after two weeks of induction in the 2N6 medium were used for gene transfer experiments. For agroinfection, modifications of the method reported by Durga Sahithya et al. [32] were implemented. Embryogenic calli were preconditioned in 2N6 medium supplemented with 100 μM acetosyringone for 60 min with gentle agitation (60–70 rpm). After this stage, calli were transferred to the bacterial culture, and the mixture was incubated for 30 min at 60–70 rpm. After infection, calli were transferred to 2N6 medium for cocultivation for 72 h at 27 ± 1 °C under a 16 h/8 h photoperiod. After coculture, calli were washed four times with 2N6 medium, with one additional wash using 2N6200cc medium. Calli were cultured in 2N6200cc for two weeks under the same conditions as before and then subjected to selection using 2N6200cc supplemented with hygromycin 25 mg/L (MilliporeSigma) for four weeks. Healthy calli were transferred to 2N6200cc supplemented with 10 mg/L hygromycin for two weeks. Calli showing active growth were transferred to regeneration medium N6R200cc. From these cultures, buds were selected and transferred to N6F medium for whole-plant generation.

### 4.5. Transgenic Condition Evaluation

-*GFP* expression. Explants were evaluated for *GFP* expression via epifluorescence microscopy using an Axio Lab.A1 microscope using the LED module at 470 nm (B) 423052-9573-000 (Zeiss, Oberkochen, Germany). The images were captured using a Cannon EOS Rebel T3 system with a GFP filter.-RNA isolation. A modified Trizol (ThermoFisher Scientific) method was used for isolating RNA from transformed embryos, as described by Azizi et al. [33]. The resulting RNA pellets were air dried for 10 min and dissolved in 30 mL of diethylpyrocarbonate-treated water. For gene expression analyses, residual genomic DNA was removed via DNase I treatment (MilliporeSigma), following the manufacturer’s procedures.-Reverse transcription: PCR detection of GFP transcripts. One microgram of total RNA was reverse transcribed using SuperScript II Reverse Transcriptase (Invitrogen, Waltham, MA, USA), following the manufacturer’s instructions. Approximately 10 ng of the synthesized cDNA was used for PCR detection of the GFP transgene based on the manufacturer’s procedures using the GFP primers (eGFP-nst1-fw CACATGAAGCAGCACGACTT and eGFP-nst1-rv AGTTCACCTTGATGCCGTTC), which amplified a 265 bp fragment. The following cycle conditions were used: 95 °C for 3 min 1 cycle, 95 °C for 30 s, 58 °C for 30 s, 72 °C for 1 min 35 cycles, and a final extension at 72 °C for 5 min, using a Bioer GeneExplorer Thermal Cycler (Bioer Technology Co., Hangzhou, China). Amplified products were electrophoresed on 1.5% (*w/v*) agarose gel containing 0.5 mg/L SerRed (Servicebio Technology, Wuhan, China) and visualized using an eBL100 Transilluminator (Eastwin Scientific Equipments Inc. Ltd., Suzhou, China).

### 4.6. Statistical Analysis

The data were analyzed for row statistics with ANOVA (two-way) and tested for significant differences (Tukey’s) using the statistical software package Prisma 10.0.2(232) (GraphPad Software Inc., Boston, MA, USA). Significant differences in embryo germination were calculated for N6R vs. N6RN using a paired t test, assuming individual variance for each genotype. Multiple comparisons with a set *p*-value threshold following the Holm–Šídák method and alpha = 0.05. The significance (95% probability level) of the difference between means is directly reported. The values were averaged over three (embryo germination) and four (GFP emitting) replicates. Original data and row statistics for each analysis are provided in Appendix A.

## Figures and Tables

**Figure 1 plants-13-00416-f001:**
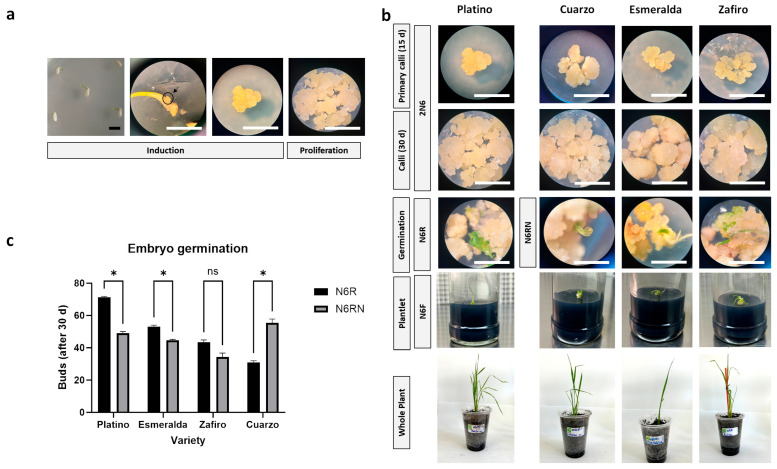
Somatic embryogenesis and regeneration in temperate japonica rice varieties Platino, Cuarzo, Esmeralda, and Zafiro. (**a**) Embryogenic calli induction and proliferation achieved through germinating seeds, isolating the zygotic embryos (arrow), and culturing these structures in 2N6 induction medium. Under these conditions, somatic embryos for all varieties reached a strong proliferative phase. (**b**) A proliferative loop was achieved between primary embryo cell masses (15 d) and embryogenic callus (30 d) with cultivation in 2N6. Calli germination was achieved via culturing embryogenic masses in N6R medium for Platino or N6RN medium for Cuarzo, Esmeralda, and Zafiro. From these cultures, active budding was obtained (“Germination”, green areas), which were isolated and cultured in N6F medium (“Plantlet”). Completely formed plants from these buds are shown at the bottom of the figure (“Whole Plant”). (**c**) Embryo germination shown as means plus standard error per genotype, including the significance (95% probability level) of the difference between means using asterisk and ns for positive or negative below-threshold results, respectively. The values were averaged over three replicates. In (**a**,**b**), bars represent 1 mm.

**Figure 2 plants-13-00416-f002:**
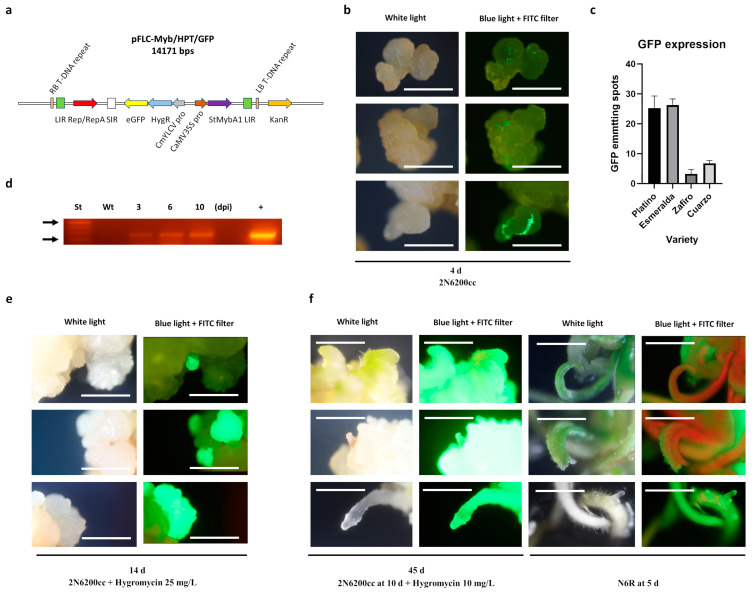
*Agrobacterium-*mediated gene transfer of rice somatic embryos. (**a**) Geminivirus-based T-DNA vector (pFLC-*Myb*/*HPT*/*GFP*) expressing green fluorescent protein (GFP), *StMyb*A1, and hygromycin resistance marker genes was used in these *Agrobacterium* EHA105-mediated gene transfer experiments. (**b**) Bacterial infections were performed at 15 d of primary calli and embryo masses being cultured in 2N6 medium. GFP-emitting spots were observed in the calli as soon as 4 d after infection. (**c**) Means of *GFP* expressing spots among varieties for the response to the gene transfer process. (**d**) Molecular detection of the *gfp* transcript in Platino as early as 3 d after infection. (**e**) Cell masses showing stable *GFP* expression were kept in this medium supplemented with cefotaxime and carbenicillin. (**f**) Calli germination and structure formation were achieved under hygromycin selection, allowing for plantlet generation in N6F medium. In a: LIR, *Large intergenic region* from the *Bean yellow dwarf virus* (*BeYDV*); Rep/RepA, geminivirus replicase gene from *BeYDV*; SIR, *Small intergenic region* from *BeYDV*; e*GFP* and HygR, *Green fluorescent protein* gene fused to the *hygromycin phophotransferase* gene; CmYLCV pro, *Cestrum yellow leaf curling virus* promoter; CaMV35S pro, *Cauliflower mosaic virus* 35S promoter; StMybA1, potato *StMybA*1 gene; Kan, *Neomycin phosphotransferase* gene. In c, means of *GFP* expressing spots are plotted as means plus standard error over four replicates. In d, St, 2 log molecular weight ladder (New England Biolabs, Ipswich, MA, USA); arrows indicate 100 (lower) and 500 (upper) bps; dpi, days post-agroinfiltration; +, pFLC-*Myb*/*HPT*/*GFP*. White bars represent 1 mm.

## Data Availability

The authors confirm that the data supporting the findings of this study are available within the article and its Appendix A.

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
