# Peer review of "Somatic Embryogenesis and Agrobacterium-Mediated Gene Transfer Procedures in Chilean Temperate Japonica Rice Varieties for Precision Breeding"

_plants, 2024, doi:10.3390/plants13030416_

Round 1

Reviewer 1 Report

Comments and Suggestions for Authors

The authors report that several varieties in rice showed different responses to hormonal conditions during somatic embryogenesis and regeneration processes. The results may include valuable information for crop improvement, but the results reported in this manuscript seem too primitive to discuss on biological mechanisms or usefulness of the authors’ findings. More logical and thorough consideration of the obtained results is apparently required.

Minor points

Line 16-17: Relationship between “Advanced biotechnological tools” and “new technologies”?

Line 47; The first >> The former?

Line 56, “different genetic engineering pipelines fuse both procedures”: Meaning unclear.

Line 69, allowed for: Is this correct?

Figures 1a and 1b: Show scale bars.

Figure 1b, bottom: Legends for the bottom four pictures are lacking.

Figure 1c: What is “Sprouting areas? What is the unit of measure for them? What do vertical bars indicate; SD, SE, or other values? In addition, the way in which P values are indicated seems odd; is this presentation style normal? Check the name of media; NR6 or N6R, NR6N or N6RN?

Line 100: What are “pipelines”?

Line 106: What are “early developed embryos”?

Line 107: What are “comparative infections”?

Line 110, (Figure 2c, lower): Bar graph above and the image of electrophoresis below include different kinds of information; the reviewer do not understand the reason why these are compiled in a single unit.

Line 132, “First obtained the species from leaves”: Meaning unclear.

Line 138, “Interestingly”: Why interesting? What does the interesting difference mean?

Line 158, “These methods are highly genotype-dependent”; What does it mean?

The reviewer does not understand how the media (or hormones) were selected and used in this study.

Comments on the Quality of English Language

It should be checked.

Author Response

Response to R#1:

-The current version of the manuscript has been edited by the MDPI´s editing service.

-Modifications according to Reviewer´s suggestion have been made and will be found in yellow in the text. Considering the comments, we have included a new paragraph in the Discussion section according to the level of results obtained. This means, the relevance of the tissue culture procedure, somatic embryogenesis itself and the factors involved in gene transfer in these genotypes, considering the state of the art, are now included. New references were added. All this considering a limitation in length due to the “Communication” format by the Journal.

-Amendments to minor points was carried out, including all the points indicated by the Reviewer.

Details:

Line 16-17: Relationship between “Advanced biotechnological tools” and “new technologies”?

OK, clarified.

Line 47; The first >> The former?

OK, amended.

Line 56, “different genetic engineering pipelines fuse both procedures”: Meaning unclear.

OK, clarified.

Line 69, allowed for: Is this correct?

OK, clarified.

Figures 1a and 1b: Show scale bars.

Done

Figure 1b, bottom: Legends for the bottom four pictures are lacking.

Done, amended.

Figure 1c: What is “Sprouting areas? What is the unit of measure for them? What do vertical bars indicate, SD, SE, or other values? In addition, the way in which P values are indicated seems odd; is this presentation style normal? Check the name of media; NR6 or N6R, NR6N or N6RN?

OK, improved and clarified in captions.

Line 100: What are “pipelines”?

OK, amended.

Line 106: What are “early developed embryos”?

OK, clarified.

Line 107: What are “comparative infections”?

OK, clarified.

Line 110, (Figure 2c, lower): Bar graph above and the image of electrophoresis below include different kinds of information; the reviewer do not understand the reason why these are compiled in a single unit.

OK, Figure divided.

Line 132, “First obtained the species from leaves”: Meaning unclear.

OK, clarified.

Line 138, “Interestingly”: Why interesting? What does the interesting difference mean?

OK

Line 158, “These methods are highly genotype-dependent”; What does it mean?

OK

The reviewer does not understand how the media (or hormones) were selected and used in this study.

Consistent citation of the mother paper [4] has been included in Methods, and all the media is detailed in Supplementary Table S1.

Comments on the Quality of English Language

It should be checked. Done.

Reviewer 2 Report

Comments and Suggestions for Authors

i have not any suggestion to the authors. 

Author Response

We thank you the effort invested making this manuscript improved. Due to comments from additional reviewers, we have prepared a new version of the manuscript which considered all the points raised by them. 

Also, as suggested, the current version of the paper was submitted to the MDPI Editing Office in order to generate an clearer and improved version of the work.

Reviewer 3 Report

Comments and Suggestions for Authors

This is a very interesting study, but the author’s description of the materials and methods section cites few references. Are the relevant methods original to the author? If not, please add relevant references and the description should be more detailed. Secondly, in the materials and methods section, the statistical analysis is relatively simple. The author should describe it in detail and provide the corresponding original data as an attachment, so as to directly prove the reliability of the data.
Other minor issues are as follows:
1. Line 21; What does N6R mean?
2. Line 22; What does N6RN mean?
3. Line 28; What does 3 d mean?
4. Line 47; a few more references need to be added.
5. lines 68, 73, 74, 76. What do 7 d, 15 d, 30 d, and 15 d mean?
6. Figure 2a is not clear and needs to be modified.
7. Line 184; Arabidopsis should be in italics.
8. The reference format needs to be modified.

Comments on the Quality of English Language

Moderate editing of English language required

Author Response

Response to R#3.

  • The current version of the manuscript has been edited by the MDPI´s editing service.
  • New references have been included in this reviewed version of the manuscript. Special emphasis was focused on Methods to clarify the somatic embryogenesis procedures, including reference to the mother reference [4]. Additionally, Supplementary Tables (S1 and new S2) include specifics for media (S1) and results (S2; raw data and statistical processing of them).
  • Results can be found in Supplementary Table S2.
  • New paragraphs have been included (in yellow) in order to clarify the points that were found confusing in the previous version of the work.

Amendments to minor points was carried out, including all the points indicated by the Reviewer.

  1. Line 21; What does N6R mean?
  2. Line 22; What does N6RN mean?

Name for these media has been the same as used in ref #4. As mentioned, cited in Methods section as much as needed.

  1. Line 47; a few more references need to be added.

New references at different points of the manuscript were included. All highlighted in yellow fonts.

  1. Figure 2a is not clear and needs to be modified.

This Figure has been improved.

Regarding the use of “d”, in general this abbreviation is used for “day”. We do not have problems changing them in the text, however, we asked to the editing office for this particular and it was recommended keeping this format.

Round 2

Reviewer 3 Report

Comments and Suggestions for Authors

The author has made modifications as required and needs to check the grammatical format before considering accepting it.

Comments on the Quality of English Language

Minor editing of English language required

Author Response

These authors thank to the reviewer for contributing to the improvement of this work. Based on R2 (from reviewer #3), we had a double checking process (personally and MDPI´s editing service (again)); and after these rounds, we are now unable to make further changes to the work at this stage as the English is already grammatically correct. I´m attaching the letter just received from MDPI´s service (today) and the previous editing certificate (for the R1 file).
